# Exploring the Regulators of Keratinization: Role of BMP-2 in Oral Mucosa

**DOI:** 10.3390/cells13100807

**Published:** 2024-05-09

**Authors:** Xindi Mu, Mitsuaki Ono, Ha Thi Thu Nguyen, Ziyi Wang, Kun Zhao, Taishi Komori, Tomoko Yonezawa, Takuo Kuboki, Toshitaka Oohashi

**Affiliations:** 1Department of Molecular Biology and Biochemistry, Okayama University Graduate School of Medicine, Dentistry and Pharmaceutical Sciences, Okayama 700-8558, Japan; pfmr6bh2@s.okayama-u.ac.jp (X.M.); thuharhm@gmail.com (H.T.T.N.); wangziyi@s.okayama-u.ac.jp (Z.W.); pgn309az@s.okayama-u.ac.jp (K.Z.); tomoy@okayama-u.ac.jp (T.Y.); oohashi@okayama-u.ac.jp (T.O.); 2Department of Oral Rehabilitation and Implantology, Okayama University Hospital, Okayama 700-8558, Japan; kuboki@md.okayama-u.ac.jp; 3Department of Oral Rehabilitation and Regenerative Medicine, Okayama University Graduate School of Medicine, Dentistry and Pharmaceutical Sciences, Okayama 700-8525, Japan; taishi.komori@nih.gov; 4Skeletal Biology Section, National Institute of Dental and Craniofacial Research, Department of Health and Human Services, National Institutes of Health, Bethesda, MD 20892, USA

**Keywords:** cell differentiation, epithelia, growth factor(s), bioinformatics, extracellular matrix (ECM), mucocutaneous disorders

## Abstract

The oral mucosa functions as a physico-chemical and immune barrier to external stimuli, and an adequate width of the keratinized mucosa around the teeth or implants is crucial to maintaining them in a healthy and stable condition. In this study, for the first time, bulk RNA-seq analysis was performed to explore the gene expression of laser microdissected epithelium and lamina propria from mice, aiming to investigate the differences between keratinized and non-keratinized oral mucosa. Based on the differentially expressed genes (DEGs) and Gene Ontology (GO) Enrichment Analysis, bone morphogenetic protein 2 (BMP-2) was identified to be a potential regulator of oral mucosal keratinization. Monoculture and epithelial–mesenchymal cell co-culture models in the air–liquid interface (ALI) indicated that BMP-2 has direct and positive effects on epithelial keratinization and proliferation. We further performed bulk RNA-seq of the ALI monoculture stimulated with BMP-2 in an attempt to identify the downstream factors promoting epithelial keratinization and proliferation. Analysis of the DEGs identified, among others, *IGF2*, *ID1*, *LTBP1*, *LOX*, *SERPINE1*, *IL24*, and *MMP1* as key factors. In summary, these results revealed the involvement of a well-known growth factor responsible for bone development, BMP-2, in the mechanism of oral mucosal keratinization and proliferation, and pointed out the possible downstream genes involved in this mechanism.

## 1. Introduction

The keratinized oral mucosa plays critical roles not only in the protection against mechanical, chemical, and biological stimuli [1] but also in the maintenance of healthy periodontal or peri-implant tissues [2,3]. Keratinization is a process of differentiation in which keratinocytes start from a post-germinative state in the basal layer and go through a fully differentiated state characterized by hardened cells filled with keratin protein and lack of nucleus in the outermost layer [4]. The mucous membrane that lines within the oral cavity is termed oral mucosa and its degree of keratinization varies depending on the location and function.

In keratinized oral mucosa, the epithelium stratifies into four layers (stratum basale, stratum spinosum, stratum granulosum, and stratum corneum), while on the other hand, it consists of three layers (stratum basale, stratum filamentosum, and stratum distendum) in non-keratinized tissue [1]. The distribution of keratins (KRTs) is recognized as a specific indicator of the differentiation and proliferation states in epithelium. Along with microfilaments and microtubules, KRTs form a structural framework within epithelial cells [5]. In the stratified epithelia, the mitotically active cells localized in the basal layer express KRT5/KRT14, and the cells exhibiting a less proliferative potential are KRT15-positive. The hyperproliferation markers, KRT6/KRT16, followed by the early differentiated markers, KRT1/KRT10, are expressed in the suprabasal layers of cornifying epithelia-like hard palatal and gingival mucosa. On the other hand, in non-keratinized epithelia KRT4 and KRT13 are the mainly expressed markers [4,6,7].

The connective tissue underneath the oral epithelium, known as lamina propria, contains collagen fibers, elastic fibers, blood vessels, nerves, sebaceous follicles, salivary glands, fibroblasts, and immune cells [8]. Lining between the epithelium and lamina propria, there is a sheet-like extracellular matrix (ECM), termed basement membrane, that plays an important role in the maintenance of tissue homeostasis [9]. It is generally believed that the epithelial–mesenchymal interaction is essential in determining the fates of epithelial cells by producing various growth factors [10]. However, the overall differences between the keratinized and non-keratinized mucosa and the mechanisms mediating the keratinization of oral mucosal epithelium remain unclear.

Bone morphogenetic protein 2 (BMP-2), a member of the transforming growth factor β (TGF-β) superfamily and a well-known factor modulating bone development, serves as a multifunctional cytokine synthesized not only by skeletal but also various extraskeletal cells [11]. In endothelial progenitor cells (EPCs), BMP-2 significantly promotes capillary tube formation and consequently triggers angiogenesis [12]. Additionally, BMP-2 has been shown to induce differentiation in various cell types, including cutaneous keratinocytes as well as stomach mucosa epithelial cells [13,14].

In this study, in an attempt to elucidate the mechanisms involved in the keratinization of oral mucosa, we performed bulk RNA-seq of laser-dissected keratinized and non-keratinized mucosa. The results identified genes associated with ossification and extracellular matrix as the differentially expressed genes (DEGs). Among them, BMP-2 was identified as a potential gene inducing epithelial differentiation. Subsequent in vitro experiments confirmed the important roles of BMP-2 in promoting the keratinization of mucosa epithelium.

## 2. Materials and Methods

### 2.1. Animals

The mice (8-week-old, male, C57BL/6 J) were purchased from CLEA Japan (Tokyo, Japan) and maintained on 12 h light/dark cycle (7 a.m.–7 p.m.) with ad libitum access to food and water. Animals were treated according to the guidelines of Okayama University Animal Research. This study has been approved by the Okayama University Animal Research Committee (OKU-2022383) and complies with the ARRIVE guidelines. Non-fixed freshly excised tissues were used for further analysis unless otherwise noted.

### 2.2. Laser Microdissection (LMD)

Palatal and buccal mucosa were collected from 3 groups of mice (10 mice/group) and immediately submerged in RNALater^®^ Solution (Life Technologies, Vilnius, Lithuania). The method for preparing frozen specimens followed the same procedures as the histological analysis described in this study. Slices at a thickness of 10 μm were transferred on PEN Membrane Slides (Leica Microsystems, Wetzlar, Germany) and dehydrated in 100% ethanol for 3 min. To facilitate the visualization of tissue morphology, a toluidine blue stain was applied to the samples and washed with RNase-free water. After drying on ice, the slides were ready for laser microdissection. Epithelial and mesenchymal layers from the buccal and palatal tissues were isolated separately under Leica Microdissection system LMD7000 (Leica Microsystems, Wetzlar, Germany) into the collection tubes containing Buffer RLT Plus (RNeasy Plus Micro Kit; QIAGEN, Hilden, Germany) and stored at −80 °C before proceeding with the RNA extraction.

### 2.3. Libraries Preparation and Sequencing

Total RNA from samples of LMD was isolated and purified using the RNeasy Plus Micro Kit (Qiagen, S.A., Courtaboeuf, France) and treated with DNase (Life Technologies, Carlsbad, CA, USA) according to the standard procedures. RNA quality and concentration were measured by Tapestation 2200 (Agilent Technologies Inc., Santa Clara, CA, USA) with High Sensitivity RNA ScreenTape (Agilent Technologies, Inc., Santa Clara, CA, USA). Then the SMART-Seq Stranded Kit (Takara Bio USA, Inc., Mountain View, CA, USA) was used to generate indexed cDNA libraries, and the purified libraries were validated by Tapestation 2200 (Agilent Technologies Inc., Santa Clara, CA, USA) with High Sensitivity D5000 ScreenTape (Agilent Technologies, Inc., Santa Clara, CA, USA). The sequencing was performed on an Illumina NovaSeq 6000 Sequencer (Illumina, San Diego, CA, USA).

The PureLink RNA Mini Kit (Life Technologies, Carlsbad, CA, USA) was utilized to isolate total RNA from the cells in ALI culture, and all the steps for purification were carried out in accordance with the provided user manual. For the generation of cDNA libraries, total RNA was input following the instructions of SMART-Seq HT PLUS Kit (Takara Bio USA, Inc., Mountain View, CA, USA). Then the libraries were sequenced using an Illumina NovaSeq 6000 Sequencer (Illumina, San Diego, CA, USA).

RNA-seq data were deposited in the NCBI SRA database, accession numbers PRJNA1054210 and PRJNA1054335.

### 2.4. Bioinformatic Analysis of Bulk RNA-Seq Data

After conducting quality trimming using Trim Galore v0.6.10, the reads were mapped to the genome with STAR aligner 2.7.10b [15] against GRCm39 (LMD samples) or GRCh38 (ALI samples) reference genomes. The de nova transcripts were generated and annotated by StringTie v2.2.1 and SQANTI3 v5.0 [16,17]. All samples were then re-quantified at the transcript level using Salmon v1.9.0 [18], and additional quality filtering was performed using isoformSwitchAnalyzeR v1.17.05 [19] based on the mapping results from Salmon. The significant differentially expressed genes (DEGs) were detected with DESeq2 v1.31.16 [20]. DEGs were defined as False Discovery Rate (FDR) < 0.05 and |log2Fold Change| > 1. Over Representation Analysis was conducted using the Gene Ontology (GO) databases and Kyoto Encyclopedia of Genes and Genomes (KEGG) database by clusterProfiler v4.6.0 [21].

The functional protein association networks generated by STRING v11.5 [22] were used to predict the interactions between the encoded proteins of DEGs in BMP-2 vs. Control ALI monoculture samples.

### 2.5. Cell Culture

Human buccal epithelial squamous carcinoma cells (TR146, DS Pharma Biomedical Co., Ltd., Osaka, Japan) were grown in F-12 Nutrient Mixture (Ham’s F-12) medium (Life Technologies, Carlsbad, CA, USA) containing 10% fetal bovine serum (FBS; Sigma-Aldrich, St. Louis, MO, USA) and 2 mM L-glutamine (Sigma-Aldrich, Gillingham, UK). Human oral mucosa fibroblasts (hOMFs, CellResearch Corporation Pte Ltd., Singapore) were cultured in Dulbecco’s modified eagle medium (DMEM, Life Technologies, Beijing, China) supplemented with 10% FBS (Sigma-Aldrich, St. Louis, MO, USA) and 2 mM L-glutamine (Sigma-Aldrich, Gillingham, UK). All the cells were incubated at 37 °C in 5% CO_2_ until 80–90% confluency before the experiments were conducted.

The air–liquid interface (ALI) cell culture was performed as described in previous studies [3,23,24] with minor modifications. Briefly, TR146 cells (1.55 × 10^5^ cells/well) were seeded into a transwell insert (0.4 μm porosity, Corning Incorporated, Corning, NY, USA) in a 24-well plate, while hOMFs (5 × 10^4^ cells/well) were seeded in the lower chamber. After being submerged in culture medium to grow until confluence, inserts with TR146 cells were either cultured alone or co-cultured with hOMFs. TR146 cells were then lifted to air–liquid interface by aspirating all the medium in the upper chamber of the culture insert and the medium was replaced in the lower chamber by differentiation medium with (*n* = 3) or without (*n* = 3) E-rhBMP-2 (250 ng/well, Osteopharma Inc., Osaka, Japan). After 12 days of differentiation, total RNA from the cells was isolated for RNA expression or RNA-seq analyses. Alternatively, the cells were fixed in 4% paraformaldehyde (PFA) for further immunostaining analysis.

### 2.6. RT-qPCR

The total RNA purification for the ALI culture samples was performed as previously outlined in this study. Two-step reverse transcription quantitative PCR was conducted with iScript cDNA synthesis kit (Bio-Rad, Hercules, CA, USA) and KAPA SYBR FAST qPCR Master Mix (KAPA BIOSYSTEMS, Cape Town, South Africa) for reverse-transcription and gene expression analysis. As an internal control, the expression levels of ribosomal protein S29 were utilized, and the relative quantification of gene expression was computed using the 2^−ΔΔCt^ method.

Nucleotide sequences of primers for target genes are shown in Appendix A.

### 2.7. Histological Analysis

Excised heads from mice (*n* = 3) were immediately fixed with 4% paraformaldehyde (PFA, Nacalai Tesque, Kyoto, Japan) for 3 days and decalcified in Morse Solution (FUJIFILM Wako Pure Chemical Corporation, Osaka, Japan) at room temperature within one week following neutralization with 5% sodium sulfate solution (FUJIFILM Wako Pure Chemical Corporation, Osaka, Japan). The paraffin sections were stained with standard hematoxylin and eosin (HE) for histological observation. The collagen fibers were represented after tissue staining with Masson’s trichrome technique.

Immunofluorescence assay was performed as described previously [3]. Briefly, samples from ALI culture (*n* = 3) and mice (*n* = 3) were embedded with super cryoembedding medium (SCEM, SECTION-LAB Co. Ltd., Yokohama, Japan) and frozen completely in cooled hexane. Then frozen sections at thickness of 7 μm were made with adhesive films (SECTION-LAB Co. Ltd., Yokohama, Japan) using the cryostat machine (Leica Biosystems Nussloch GmbH, Nussloch, Germany), dipped in cold acetone or methanol for 10 min. After treatment with a blocking solution for 1 h at room temperature, the samples were incubated with primary antibody KRT10 (ab76318, Abcam, Cambridge, UK, 1:500), KRT14 (ab181595, Abcam, 1: 3000) or Ki67 (ab16667, Abcam, 1:500) overnight at 4 °C. The secondary antibody, Goat anti-rabbit IgG(H+L) Alexa Flour Plus 488 (A32731, Invitrogen, Gaithersburg, MD, USA, 1:400) or Goat anti-rabbit IgG(H+L) Alexa Flour Plus 647 (A32733, Invitrogen, Gaithersburg, MD, USA, 1: 500) were used to cover each sample and simultaneously underwent nuclear staining with DAPI (Life technologies, Gaithersburg, MD, USA) for 1 h at room temperature. The images of stained sections were observed and captured by BZ-700 fluorescence microscope (Keyence, Osaka, Japan). Immunofluorescence samples were quantified by the percentage of positive cells out of the total cell counts.

### 2.8. Statistical Analysis

Significance values for the RT-qPCR and immunofluorescence data were calculated with two-tailed unpaired *t*-test (GraphPad Software v9.1.1, Inc., San Diego, CA, USA). All experiments were performed in triplicate and the significance was called when *p*-value ≤ 0.05.

## 3. Results

### 3.1. Extracellular Matrix and Ossification-Associated Genes Were Candidate Regulators of Keratinization

To assess the overall gene profile of the keratinized and non-keratinized oral mucosa, the tissues were microdissected to separate the epithelium from the underlying lamina propria. The isolated epithelial and mesenchymal tissues were further subjected to RNA sequencing (Figure 1A).

The analysis revealed 1484 protein-coding DEGs in the epithelium, and 2006 DEGs in the lamina propria when comparing palatal (keratinized) to buccal (non-keratinized) mucosa (Figure 1B,C, Appendix A). Interestingly, we found 67 DEGs, including *Bmp7*, *Dapl1*, *Tesc*, *Herc4*, and *Bmp2*, that were significantly changed and expected to be associated with “differentiation”, based on Uniprot keywords (Figure 1D and Appendix A). Functional Enrichment Analysis was then carried out to identify the pathways that could be potentially relevant to keratinization. The analysis identified 905 Gene Ontology_ biological process (GO_BP) terms and 43 KEGG pathways from the upregulated DEGs, such as extracellular matrix organization (GO:0045229, GO:0030198, GO:0043062), skin development (GO:0043588), bone ossification (GO:0001503, GO:0060348) and TGF-beta signaling pathway (mmu04350) (Figure 1E and Appendix A).

During the comparison of lamina propria mesenchyme, we also observed 85 “differentiation” associated DEGs, including *Csrp3*, *Sfrp2*, *Bmp7*, and *Bmp2*, that were partially overlapped with our epithelial comparison (Appendix A). The upregulated DEGs in the mesenchyme of keratinized mucosa were enriched in 1496 GO_BP terms and 51 KEGG pathways, such as extracellular matrix organization (GO:0030198, GO0043062, GO:0045229), bone ossification (GO:0001503, GO:0001649), cartilage development (GO:0061448, GO:0051216, GO:0048705), and TGF-beta signaling pathway (mmu04350) (Figure 1F and Appendix A). These results suggest that the differences between keratinized and non-keratinized mucosa may potentially be attributed to the components of ossification, extracellular matrix (ECM), and the members of the TGF-beta family.

Furthermore, it was interesting to note that within the top 5 significant GO_BP terms from both the epithelial and the mesenchymal comparison, a few genes, e.g., *Col2a1*, *Lox*, *Col13a1*, *Ddr2*, *Bmp2*, *Sfrp2*, *Col1a2*, *Npnt*, *Comp*, and *Mkx*, were found to be overlapped in multiple annotation categories associated with ossification and ECM. This finding demonstrated the potential importance of these genes in regulating the differentiation of keratinized mucosa (Figure 1G,H).

### 3.2. BMP-2 Functions as an Inducer of Keratinization and Proliferation in ALI Culture

Among the DEGs belonging to ossification and extracellular matrix, BMP-2, which plays a crucial role in regulating osteoblastogenesis and bone formation [25], was selected for further analysis. To further validate the molecular functions of BMP-2 in either epithelial or mesenchymal cells, we generated an ALI monoculture model with only the oral mucosal epithelial cells and a co-culture model with both the epithelial cells and fibroblasts.

Monocultured TR146 cells treated with BMP-2 in the ALI culture for 12 days showed a significant increase in the gene expression of *KRT10* and *IVL* (Figure 2A,B), two well-known markers for epithelial cells undergoing keratinization in the suprabasal and cornified layers. However, such enhancement of *KRT10* and *IVL* expression was not found in co-culture experiments (Figure 2C,D). In addition, the proliferation capability was investigated by analyzing the gene expression of specific markers in monocultured and co-cultured TR146. An acidic intermediate filament protein coding gene typically expressed in slow-growing epithelial cells, termed *KRT15*, was significantly decreased after treatment with BMP-2 (Figure 2E,F). Conversely, only the monocultured groups exhibited a significant increase in the proliferation markers, *KRT14* and *KRT16*, which suggested a notable elevation in their proliferative potential (Figure 2G–J).

We further determined the expression of keratinized and proliferative markers at a protein level by immunofluorescence analysis. Consistent with the significant increase in the expression of genes, the protein levels of KRT10 and KRT14 were markedly higher in the monocultured TR146 after BMP-2 treatment compared to those of the control samples. Nevertheless, BMP-2 failed to induce KRT10 or KRT14 in the co-culture groups (Figure 2K,L), which indicated its multifunction in modulating the keratinization process. Images were compared to negative controls using the secondary antibody only (Appendix A).

### 3.3. The Downstream Targets of BMP-2 Are Crucial in Regulating the Keratinization of Oral Mucosa

Once initiated, canonical BMP signaling cascade recruits and phosphorylates Smad1/5/8 (also known as Smad9), which subsequently form a gene regulation complex with Smad4. Alternatively, BMPs can also trigger a non-canonical downstream signaling pathway through a diversity of intracellular kinases [25]. To explore the potential target genes involved in BMP-2-induced keratinization, we further performed bulk RNA-seq using BMP-2 treated/untreated cells from ALI monoculture models. Our results identified 151 genes that were differentially expressed in the epithelial cells upon BMP-2 stimulation, among which we prioritized several significantly different genes associated with the TGF-beta signaling pathway (GO:0007179, GO:0071559), including *LTBP1*, *LOX*, and *ID1*. Additionally, functional enrichment analysis indicated that the upregulated genes *LOX*, *PTX3*, and *MMP1* were highly correlated to the extracellular matrix (GO:0030198, GO:0043062, GO:0045229). Furthermore, we confirmed enrichment of the biological process of wound healing (GO:0042060, GO:0061042) by upregulation of the DEGs like *SERPINE1*, *LOX*, *IL24*, and the positive regulation of epithelial cell proliferation (GO:0050679) by upregulation of a set of genes, such as *ID1* and *IGF2*. Notably, we found multiple highly significant DEGs enriched in these terms, suggesting their potential roles as the downstream targets of BMP-2 in oral mucosal epithelial cells (Figure 3A–C). Next, functional protein association networks revealed the protein-protein interaction within the encoded proteins of the top 55 most significant DEGs and suggested that the proteins of LOX, LTBP1, SERPINE1, MMP1, PTX3, ID1, IGF2, CA9, and LCN2 were inclined to interact with each other (Figure 3D). As expected from the RNA-seq data, significant enhancement in the expression of these candidate genes was validated by RT-qPCR (Figure 3E).

These results highlight the importance of the overlapping genes from enrichment analysis and the prediction of functional protein associations, as they may function as the downstream mediators of the BMP-2-induced oral mucosal keratinization/proliferation. As shown in Table 1, lysyl oxidase (LOX), latent transforming growth factor beta binding protein 1 (LTBP1), and matrix metalloproteinase 1 (MMP1) have been reported to possess the ability to induce epithelial differentiation, while serpin family E member 1 (SERPINE1), inhibitor of DNA binding 1 (ID1), and insulin-like growth factor 2 (IGF2) are activators of epithelial proliferation. Furthermore, interleukin 24 (IL24), one of the significant DEGs belonging to the GO_BP term “wound healing” (Figure 3C), has been documented to induce hyperproliferation and inhibit late differentiation in the epidermal keratinocytes.

## 4. Discussion

In a clinical setting, an “adequate” width (≥2 mm) of keratinized mucosa is required for dental implant treatment, to reduce plaque accumulation, bleeding, and mucosal recession [2]. However, it is still challenging to augment the lost keratinized mucosa with various surgical techniques because of limitations in tissue manipulation and postoperative discomfort [44]. Hence, we aim to explore the mechanisms and identify potential key factors involved in oral mucosal keratinization, offering new insights for the development of novel therapeutics for future clinical application. In this study, depending on the histological features and differentially expressed early keratinization marker KRT10 (Appendix A), palatal and buccal mucosa were chosen to represent keratinized and non-keratinized oral mucosa [4]. These tissues were then isolated for subsequent comprehensive transcriptomic analysis, which provided direct evidence of the key components associated with epithelial keratinization.

Bulk RNA-seq analysis of the keratinized and non-keratinized mucosa allowed us to compare these two types of oral mucosa and revealed highly significant GO_BP terms from the upregulated DEGs in both the epithelium and mesenchyme. Notably, ECM and ossification-related terms were identified, suggesting their potential roles responsible for the differences between the two types of mucosae (Figure 1E,F). The components of ECM, like fibronectin, chondroitin sulfate proteoglycan, and elastin differ between the dermis (skin) and lamina propria (oral mucosa), while the oral tissue has more layers of epidermal cells undergoing proliferation, but a lower tendency to be differentiated comparing with the cutaneous epidermis [45]. Consistent with these features, our histological analysis of the mucosa samples observed densely organized collagenous connective tissue fibers underneath the epithelial layer, and tightly bound to the underlying bone in palatal mucosa (keratinized mucosa) (Appendix A).

As members of the transforming growth factor β (TGF-β) superfamily, BMP-2 and BMP-7 have been widely used as osteogenic agents for their ability to induce osteoblast differentiation and stimulate bone or cartilage formation [11]. Consistent results were shown in our list of DEGs, BMP-2 and BMP-7 were identified as differentiation-related genes by sorting with Uniprot keywords (Figure 1D). However, within the top 5 significant GO terms in both epithelial and mesenchymal comparison, only BMP-2 was shown to be one of the overlapping genes from ECM and ossification-correlated terms (Figure 1G,H). Moreover, the biological involvement of BMP-2 in inducing epithelial differentiation (KRT10/IVL) is studied in cutaneous keratinocytes, accompanied by the downregulation of p53-mediated transcriptional activity [13]. Thus, we aimed to confirm the essential role of the growth factor BMP-2 during oral mucosal epithelial differentiation.

After treatment with BMP-2, monocultured TR146 cells exhibited a differentiation state with a significant increase in the expression of the keratinized markers (*KRT10* and *IVL*), and a higher proliferation capacity associated with the increase in *KRT14*, *KRT16*, *MKI67*, *and CCND1* genes (Figure 2A,B,G,H and Appendix A). Unexpectedly, BMP-2 had little impact on both keratinization and proliferation of the TR146 cells when they were co-cultured with the fibroblasts (hOMFs) (Figure 2C,D,I,J). Of note, because of the potential limitations of the ALI model, which is required to culture in a differentiation medium and tends to exhibit a significant decrease in Ki67-positive cells over 9 days [46], no significant alteration of Ki-67 was found after BMP-2 treatment. The signals were scattered and sparse in both co-culture and monoculture models (Appendix A).

The binding of BMP-2 with BMP receptor type Ia and type Ib (BMPRIa and BMPRIb) is required for the activation of downstream Smad and non-Smad signaling pathways, which contribute to the regulation of target genes including Ids, Runx2, Lox and Col1a1 [47,48]. To study the potential downstream target genes involved in BMP-2-stimulated epithelial keratinization and proliferation, we subjected the ALI monoculture samples to RNA-Seq analysis. The elevation of keratinization marker *IVL* and proliferation marker *KRT14*, as well as the diminishment of slow-growing marker *KRT15*, were verified to be the same as our RT-PCR result (Figure 2B,E,G, and Figure 3A). Interestingly, similar to the results of the enrichment analysis of laser-dissected tissues, the TGF-beta signaling pathway, cell-cell adhesion, wound healing, positive regulation of epithelial cell proliferation, and extracellular matrix organization, were enriched from the upregulated DEGs in the monoculture groups by treatment of BMP-2, indicating their functional significance (Figure 3B).

Among the top 55 significantly altered genes in ALI monoculture, *LOX*, *LTBP1*, *SERPINE1*, *ID1*, *PTX3*, *MMP1*, *IGF2*, *IL24*, and *ROBO4* were also involved in the enriched GO_BP terms of our interests (Figure 3C). Moreover, except for *IL24*, and *ROBO4*, their encoded proteins were predicted to have protein-protein interaction with each other by the analysis of functional protein association networks (Figure 3D). We then attempted to explore the roles of each of these BMP-2 responsive genes identified in the RNA-Seq of ALI monocultures.

The expression level of Lox is reported to be acutely enhanced by BMP-2 via Smad4 and Runx2 [48]. As the activation molecules to the deposition of ECM components, LOX and LTBP1 (a protein assisting in the triggering of TGF-beta) are also reported to act as stimulators of the epithelial differentiation in skin and oral mucosa [26,28,29,30]. In psoriatic skin, the epidermal keratinocytes exhibit a tendency towards hyperproliferation with elevated expression of SERPINE1 and ID1. Moreover, the induction of SERPINE1 and ID1 on proliferation is validated in oral squamous cell carcinoma and normal cutaneous keratinocytes [32,33,34,35]. Interestingly, in contrast to its upregulation after BMP-2 stimulation in our ALI cell culture, IL24 has been reported to promote the proliferation in cutaneous keratinocytes, but on the other hand, also acts as an inhibitor of the epidermal late differentiation (including FLG, but not IVL) via specifically activating STAT3, which may explain no change in this late differentiation marker in our ALI cell culture (Appendix A) following BMP-2 treatment [40,41] (Table 1). Considering their diverse activities on epithelial fates, these correlated genes downstream are likely to serve as potential regulators of keratinization and proliferation induced by BMP-2.

## 5. Conclusions

Altogether, we anticipate the potential involvement of ossification and extracellular matrix components in achieving successful terminal epithelial differentiation in the keratinized mucosa. Furthermore, these results suggest the involvement of a well-known bone development trigger element, BMP-2, in the mechanism of oral mucosal keratinization and proliferation via its downstream target genes.

## Figures and Tables

**Figure 1 cells-13-00807-f001:**
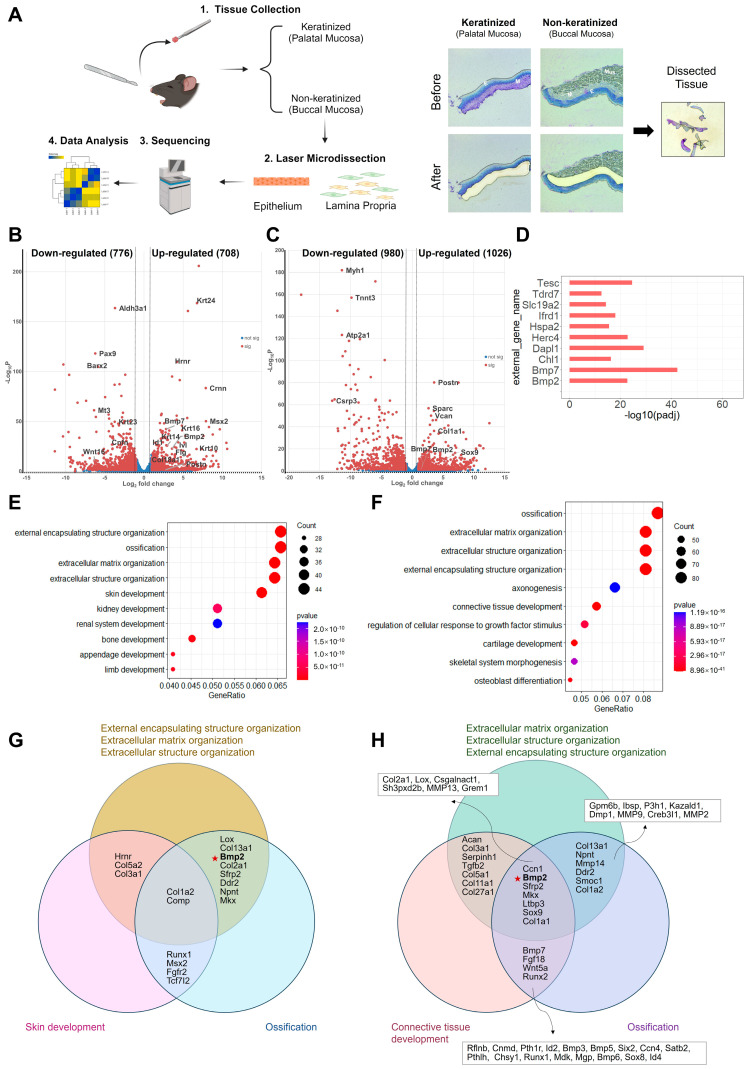
Comprehensive gene profile comparison between keratinized and non-keratinized mucosa. Epithelium and lamina propria from mucosal tissues were isolated by laser microdissection and submitted to RNA sequencing (created in BioRender) (**A**). Volcano plots showing the upregulated (right, red) and downregulated (left, red) differentially expressed genes (DEGs) in the epithelial (**B**) and mesenchymal (**C**) comparison. DEGs were defined as False Discovery Rate (FDR) < 0.05 and |log2Fold Change| > 1. The top 10 “differentiation”-associated significant DEGs between the keratinized and non-keratinized epithelium (**D**). The top 10 most significantly enriched GO_BP terms from over-representation analysis of upregulated DEGs in the epithelial (**E**) and mesenchymal (**F**) comparison. DEGs belonged to multiple annotation categories of the top 5 most significant GO_BP in the epithelial (**G**) and mesenchymal (**H**) comparison. E, epithelium; M, mesenchyme; Mus, muscle. Pentagrams indicate the DEG named bone morphogenic protein-2 (BMP-2).

**Figure 2 cells-13-00807-f002:**
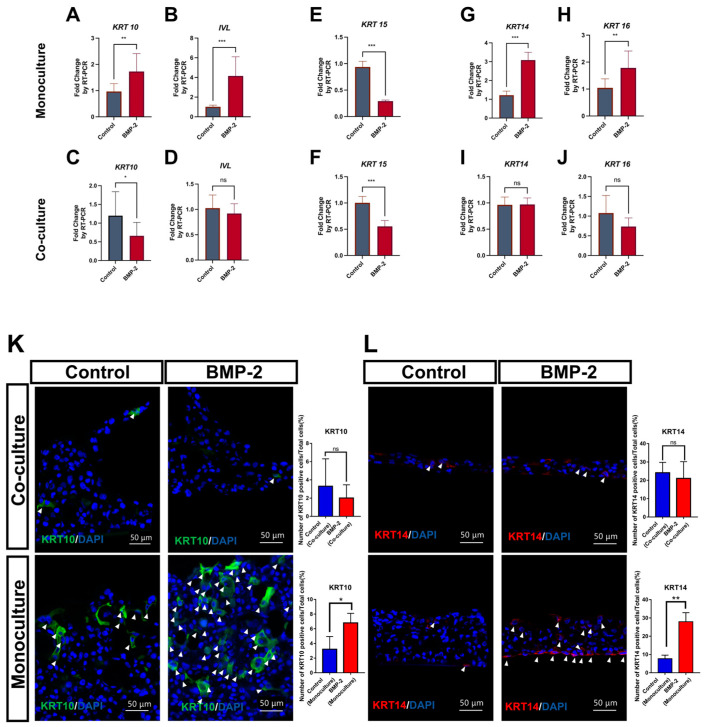
BMP-2 stimulation altered the expression of keratinized and proliferative markers in air–liquid interface (ALI) cell culture. Quantification of differentiation/proliferation-associated genes after treatment with BMP-2 in co-cultured and monocultured ALI (**A**–**J**). Immunofluorescence staining analysis of KRT10 (**K**) and KRT14 (**L**) expression after treatment with BMP-2 in co-cultured and monocultured ALI. Arrowheads indicate positive cells for KRT10 and KRT14. Scale bar: 50 μm. The graphs show the number of positive cells/total cells (%). All experiments were performed in triplicate (*n* = 3). Data are expressed as mean ± SD. ns = non-significant. * *p* ≤ 0.05, ** *p* ≤ 0.01, *** *p* ≤ 0.001, two-tailed unpaired *t* test.

**Figure 3 cells-13-00807-f003:**
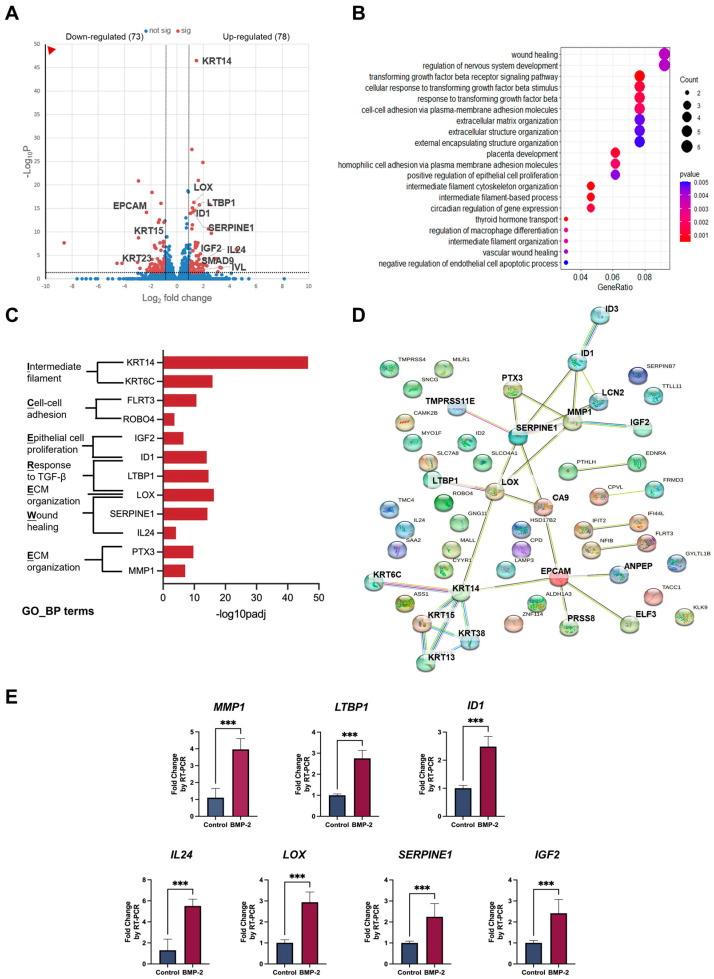
Identification of the candidate genes downstream of BMP-2 involved in epithelial keratinization and proliferation. Volcano plot showing the upregulated (right, red) and downregulated DEGs (left, red) after treatment with BMP-2 (**A**). DEGs were defined as False Discovery Rate (FDR) < 0.05 and |log2Fold Change| > 1. Dot plot expressing the top 20 most significant GO_BP terms from upregulated DEGs (**B**). The most significant DEGs enriched in the GO_BP terms (top 20, from upregulated DEGs) (**C**). Analysis of protein–protein interactions among the encoded proteins of the top 55 DEGs (**D**). Quantification of the potential downstream genes involved in BMP-2 stimulated keratinization/proliferation by RT-qPCR (*n* = 3) (**E**). Data are expressed as mean ± SD. *** *p* ≤ 0.001, two-tailed unpaired *t* test.

**Table 1 cells-13-00807-t001:** The characterization of candidate genes downstream of BMP-2.

Gene Symbol	Cell Type/Disease	Function	Reference
LOX	Oral submucous fibrosis, Oral squamous cell carcinoma	Induce deposition of collagen component.	[26,27]
	Epidermal keratinocytes	Induce epithelial differentiation.	[28]
LTBP1	Oral mucosal epithelial cells	Induce epithelial differentiation.	[29]
	Cutaneous fibroblasts	Promotes fibrillin incorporation into the extracellular matrix.	[30]
SERPINE1	Gingival wound healing	Upregulated during wound healing.	[31]
	Psoriatic skin, basal cell carcinoma	Upregulated in lesions (hyperproliferation) than normal skin.	[32]
	Oral squamous cell carcinoma	Enhance cell proliferation; Inhibit cell apoptosis.	[33]
ID1	Epidermal keratinocytes	Induce cell proliferation.	[34]
	Psoriatic skin	Upregulated in epidermis than normal skin.	[35]
MMP1	Epidermal keratinocytes	Induce cell differentiation (E-cadherin) via collagen fragmentation.	[36]
	Gingival wound healing	Upregulated during wound healing.	[31]
IGF2	Psoriasis	Induce angiogenesis.	[37]
	Anal squamous cell carcinoma, Prostate cancer	Induce cell proliferation.	[38,39]
IL24	Epidermal keratinocytes	Induce keratinocyte proliferation and hyperplasia (KRT6, KRT16); inhibit epidermal late differentiation (filaggrin, loricrin).	[40,41]
	Cutaneous wound healing	Inhibit cell migration.	[42]
ROBO4	Endothelial cell	Induce junctional adhesion molecules (ZO-1, occludin).	[43]

## Data Availability

RNA-seq data were deposited in the NCBI SRA database, accession numbers PRJNA1054210 and PRJNA1054335.

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
