# Peer review of "Exploring the Regulators of Keratinization: Role of BMP-2 in Oral Mucosa"

_cells, 2024, doi:10.3390/cells13100807_

Round 1

Reviewer 1 Report

Comments and Suggestions for Authors

Mu et. al, utilize bulk RNA sequencing of epithelium and lamina propria of oral mucosa coupled by immunofluorescence of keratinization markers to elucidate the role of BMP-2 in keratinization. The manuscript is well written but a bit descriptive lacking mechanistic insight. 

Below are my comments:

1. In the introduction section authors should provide a brief background on what’s known regarding BMP-2 role in other epithelial tissue.

2. Authors should also add the DEG numbers in the volcano plot in Figures 1 and 3 to better assist the readers regarding the magnitude of the difference. Also, add an Excel file for upregulated and downregulated genes.

3. In Figure 2 authors should use either ki67 or EdU staining to support their claim for proliferation.

4. To provide a better mechanistic insight authors should do a knockdown experiment of some of the proposed downstream target genes upon BMP-2 induction to verify its effect.

Author Response

Mu et. al, utilize bulk RNA sequencing of epithelium and lamina propria of oral mucosa coupled by immunofluorescence of keratinization markers to elucidate the role of BMP-2 in keratinization. The manuscript is well written but a bit descriptive lacking mechanistic insight.

Below are my comments:

Comment #1: In the introduction section authors should provide a brief background on what’s known regarding BMP-2 role in other epithelial tissue.

Response #1: Thank you for your suggestions. The following text has been added and modified as necessary; please see lines 64-70 of the manuscript. 

Lines 64-70:

Bone morphogenetic protein 2 (BMP-2), a member of the transforming growth factor β (TGF-β) superfamily and a well-known factor modulating bone development, serves as a multifunctional cytokine synthesized not only by skeletal but also various extraskeletal cells [11]. In endothelial progenitor cells (EPCs), BMP-2 significantly promotes capillary tube formation and consequently triggers angiogenesis [12]. Additionally, BMP-2 has been shown to induce differentiation in various cell types, including cutaneous keratinocytes as well as stomach mucosa epithelial cells [13,14].

Comment #2: Authors should also add the DEG numbers in the volcano plot in Figures 1 and 3 to better assist the readers regarding the magnitude of the difference. Also, add an Excel file for upregulated and downregulated genes.

Response #2: Thank you for your suggestions. The manuscript has been revised according to the reviewer’s suggestion. Please see the DEGs list in the Supplementary files and Figure 1 and 3.

Comment #3: In Figure 2 authors should use either ki67 or EdU staining to support their claim for proliferation.

Response #3: As suggested by the reviewer, we have confirmed the gene expression of three proliferation markers (MKI67, PCNA, CCND1) by RT-qPCR, and detected Ki67 expression at protein level by immunofluorescence analysis. A slight but significant increase in MKI67 and CCND1 gene expression was detected in the monoculture group, although there was no significant difference in ki67 staining observed in both monoculture and co-culture model. For more detail, please see line 356, lines 363-367 and supplementary Fig. S5 of the revised manuscript.

Comment #4: To provide a better mechanistic insight authors should do a knockdown experiment of some of the proposed downstream target genes upon BMP-2 induction to verify its effect.

Response #4: Thank you for your suggestions. To validate the downstream genes, in the revised manuscript we examined the effect of BMP-2 stimulation on these candidate genes by RT-qPCR (see Figure 3E). We then confirmed that the results were similar to the RNA-seq results.

We fully understand that it is important to perform knockdown and knock-in experiments to further confirm the effects of these genes, as you have suggested. However, many knock-in and knock-out experiments have already been performed and reported regarding the function of these genes in epithelial cells. For example, Provost et al. have shown that knockdown of LOX suppresses epithelial cell keratinization, and Hamajima et al. reported that ID1 transfection stimulated cell proliferation using various techniques. Therefore, we did not perform the knockdown experiments ourselves in the present study, but summarized those results in Table and discussed that these genes may play important roles in BMP-2-stimulated epithelial keratinization and proliferation.

Moreover, the ultimate goal of this study is to develop a treatment to induce the differentiation of non-keratinized gingiva to keratinized gingiva using drugs in the dental clinical setting, and our finding BMP-2 in this study was a significant step forward. Thank you for all the suggestions.

Reviewer 2 Report

Comments and Suggestions for Authors

'Exploring the Regulators of Keratinization: Role of BMP-2 in Oral Mucosa' by Mu et. al., in this study, for the first time, bulk RNA-seq analysis was performed to explore the gene expression of laser microdissected epithelium and lamina propria from mice, aiming to investigate the differences between keratinized and non-keratinized oral mucosa. Based on the differentially expressed genes (DEGs) and Gene Ontology (GO) Enrichment Analysis, bone morphogenetic protein 2 (BMP-2) was identified to be a potential regulator of oral mucosal keratinization. Monoculture and epithelial-mesenchymal cell co-culture models in Air-Liquid Interface (ALI) indicated that BMP-2 has direct and positive effects on epithelial keratinization and proliferation. The authors further performed bulk RNA-seq of the ALI monoculture stimulated with BMP-2 in an attempt to identify the downstream factors promoting epithelial keratinization and proliferation. Analysis of the DEGs identified, among others, IGF2, ID1, LTBP1, LOX, SERPINE1, IL24, and MMP1 as key factors. The results revealed the involvement of a well-known growth factor responsible for bone development, BMP-2, in the mechanism of oral mucosal keratinization and proliferation, and pointed out the possible downstream genes involved in this mechanism. My minor concern below:

1. The authors must show that Bmp2 gene is differentially expressed significantly by real time PCR between mRNA isolated from keratinized and non-keratinized mucosa and at the protein level by western blot between keratinized and non-keratinized mucosa cell lysate. 

Author Response

'Exploring the Regulators of Keratinization: Role of BMP-2 in Oral Mucosa' by Mu et. al., in this study, for the first time, bulk RNA-seq analysis was performed to explore the gene expression of laser microdissected epithelium and lamina propria from mice, aiming to investigate the differences between keratinized and non-keratinized oral mucosa. Based on the differentially expressed genes (DEGs) and Gene Ontology (GO) Enrichment Analysis, bone morphogenetic protein 2 (BMP-2) was identified to be a potential regulator of oral mucosal keratinization. Monoculture and epithelial-mesenchymal cell co-culture models in Air-Liquid Interface (ALI) indicated that BMP-2 has direct and positive effects on epithelial keratinization and proliferation. The authors further performed bulk RNA-seq of the ALI monoculture stimulated with BMP-2 in an attempt to identify the downstream factors promoting epithelial keratinization and proliferation. Analysis of the DEGs identified, among others, IGF2, ID1, LTBP1, LOX, SERPINE1, IL24, and MMP1 as key factors. The results revealed the involvement of a well-known growth factor responsible for bone development, BMP-2, in the mechanism of oral mucosal keratinization and proliferation, and pointed out the possible downstream genes involved in this mechanism. My minor concern below:

Comment #1: The authors must show that Bmp2 gene is differentially expressed significantly by real time PCR between mRNA isolated from keratinized and non-keratinized mucosa and at the protein level by western blot between keratinized and non-keratinized mucosa cell lysate. 

Response #1: Thank you for pointing this out. The authors completely agree with the reviewer. However, in this study, to identify the differences of epithelium or mesenchyme separately, we used laser microdissection to isolate these two components from oral mucosa. We cut more than 100 tissues on slides to collect the RNA for each sample. Finally, we were able to collect and purify several nanograms of RNA required for RNA sequencing. Indeed, we wanted to confirm by PCR and western blotting, but it was not technically possible from the point of view of sample collection.

Reviewer 3 Report

Comments and Suggestions for Authors

The article entitled with “Exploring the Regulators of Keratinization: Role of BMP-2 in 2 Oral Mucosa” focused on the BMP-2 induced epithelial differentiation in oral mucsa and identified the important roles of BMP-2 in promoting the keratinization of mucosa epithelium. This article is well organized and pretty readable, but I still have some concerns:

Major concern:

1.       In figure 2 A-J, authors used both monoculture model and co-culture model with both the epithelial cells and fibroblasts, and different genes expression results was observed. But it seems they collected the co-culture RNA samples with both two cell groups, so it is hard to identify whether these genes expression has been changed in mucosal epithelial cells. To elucidate this question, the authors can use transwell co-cultule model and the collect epithelial cells for analysis.

2.       In figure 3, the results suggested that LOX and LTBP1, et al. may function as the downstream mediators of the BMP-2-induced oral mucosal keratinization. Since the authors have already collected the mucosal epithelial cells RNA samples with and without BMP-2 treatment, they can use these RNA samples to identify whether these genes expression are changed, which can increase the integrity to this article.

3.       In linen 389, there is figure S5 that has never been cited in the article text, please note that all the figures should been cited in the article text.

4.       In graphical abstract and figure 1A, it seems that the authors have produced these figures with some drawing software, if yes, they should obtain use copyright.

Minor concern:

1.       In line 112 and 182, all the software that used in bioinformatic or statistical analysis should be annotated with exact version.

2.       In figure 2 and figure S4, figure legend should contain the exact n values in all the bar chart.

3.       In line 239-248, authors should refer to the graphs in figure 2 in sequentially alphabetical order.

4.       The Table 1 is not clear to read, the authors should change the front and line spacing to make the table clear.

Author Response

The article entitled with “Exploring the Regulators of Keratinization: Role of BMP-2 in 2 Oral Mucosa” focused on the BMP-2 induced epithelial differentiation in oral mucsa and identified the important roles of BMP-2 in promoting the keratinization of mucosa epithelium. This article is well organized and pretty readable, but I still have some concerns:

Major concern:

Comment #1: In figure 2 A-J, authors used both monoculture model and co-culture model with both the epithelial cells and fibroblasts, and different genes expression results was observed. But it seems they collected the co-culture RNA samples with both two cell groups, so it is hard to identify whether these genes expression has been changed in mucosal epithelial cells. To elucidate this question, the authors can use transwell co-cultule model and the collect epithelial cells for analysis.

Response #1: Thank you for your kind suggestions. To collect the RNA from only epithelial cells, the Air-liquid interface system (using transwell) was used for both the monoculture and co-culture model in our study. Please see “Materials and Methods_2.5. cell culture” (Lines 142-153) for more information.

Lines 142-153:

The Air-Liquid Interface (ALI) cell culture was performed as described in previous studies [3,23,24] with minor modifications. Briefly, TR146 cells (1.55×105 cells/well) were seeded into a transwell insert (0.4 μm porosity, Corning Incorporated, Corning, NY, USA) in a 24-well plate, while hOMFs (5×104 cells/well) were seeded in the lower chamber. After being submerged in culture medium to grow until confluence, inserts with TR146 cells were either cultured alone or co-cultured with hOMFs. TR146 cells were then lifted to air-liquid interface by aspirating all the medium in the upper chamber of the culture insert and the medium was replaced in the lower chamber by differentiation medium with (n=3) or without (n=3) E-rhBMP-2 (250 ng/well, Osteopharma Inc., Osaka, Japan). After 12 days of differentiation, total RNA from the cells was isolated for RNA expression or RNA-seq analyses. Alternatively, the cells were fixed in 4% paraformaldehyde (PFA) for further immunostaining analysis.

Comment #2: In figure 3, the results suggested that LOX and LTBP1, et al. may function as the downstream mediators of the BMP-2-induced oral mucosal keratinization. Since the authors have already collected the mucosal epithelial cells RNA samples with and without BMP-2 treatment, they can use these RNA samples to identify whether these genes expression are changed, which can increase the integrity to this article.

Response #2: We agree with the reviewer’s suggestions. Accordingly, we have confirmed the gene expression of these potential downstream genes by RT-qPCR and added these results into Fig. 3E and lines 294-295. 

Comment #3: In linen 389, there is figure S5 that has never been cited in the article text, please note that all the figures should been cited in the article text.

Response #3: Thank you for pointing this out. The Figure S5 (Fig. S3 in the revised manuscript) has been cited. We have also changed the order of the supplementary figures according to their appearance in the revised manuscript. Please see lines 262, 325, 341, 397, 409-411 and Fig. S3-6 in supplementary files.

Comment #4: In graphical abstract and figure 1A, it seems that the authors have produced these figures with some drawing software, if yes, they should obtain use copyright.

Response #4: Thank you for your suggestion. We used Biorender to create both the graphical abstract and Figure 1A. The publication license has been uploaded and we also mentioned the usage of such the software in the Figure legend (line 229) and Acknowledgments (lines 425-426).

Minor concern:

Comment #1: In line 112 and 182, all the software that used in bioinformatic or statistical analysis should be annotated with exact version.

Response #1: Thank you for your concern. We have added the exact version of each software used in the manuscript. Please see lines 119-131 and 189 in the modified manuscript.

Comment #2: In figure 2 and figure S4, figure legend should contain the exact n values in all the bar chart.

Response #2: The manuscript has been revised as suggested. Please see the legends for Fig. 2, Fig. S5 and Fig. S6

Comment #3: In line 239-248, authors should refer to the graphs in figure 2 in sequentially alphabetical order.

Response #3: The manuscript has been revised as suggested. We have changed the order of Fig. 2 and the parts that refer to them in the manuscript. Please see lines 249-255, lines 356-358 and line 375.

Comment #4: The Table 1 is not clear to read, the authors should change the front and line spacing to make the table clear.

Response #4: We have re-arranged the table, according to the reviewer’s suggestion. Please see line 316. We hope these revisions and the improved table will be satisfactory.

Reviewer 4 Report

Comments and Suggestions for Authors

This study evaluated the differential gene expression between keratinised and non-keratinised oral epithelium. It is a well written manuscript with clearly presented methodology. It would be interesting if the authors would further discuss possible clinical implications of this study in the discussion section

Author Response

Comment #1: This study evaluated the differential gene expression between keratinised and non-keratinised oral epithelium. It is a well written manuscript with clearly presented methodology. It would be interesting if the authors would further discuss possible clinical implications of this study in the discussion section

Response #1: Thanks for the very valuable comments. According to your suggestion, we discussed the possible clinical application of our works, and added to the new revised manuscript. Please see lines 318-324.

Lines 318-324:

In a clinical setting, an “adequate” width (≥ 2 mm) of keratinized mucosa is required for dental implant treatments, to reduce plaque accumulation, bleeding and mucosal recession [2]. However, it is still challenging to augment the lost keratinized mucosa by various surgical techniques because of limitations in tissue manipulation and postoperative discomfort [44]. Hence, we aim to explore the mechanisms and identify potential key factors involved in oral mucosal keratinization offering new insights for the development of novel therapeutics for future clinical application.

Round 2

Reviewer 1 Report

Comments and Suggestions for Authors

The authors have sufficiently addressed the comments. 

Reviewer 2 Report

Comments and Suggestions for Authors

I agree with the authors justification of not being able to perform real time PCR on the samples due to minute amounts of RNA recovered. I recommend the manuscript for publication.

Reviewer 3 Report

Comments and Suggestions for Authors

The authors have addressed all my concerns.